

# Correlated evolution of sternal keel length and ilium length in birds

Tao Zhao[1],  Di Liu[2,3,4] and  Zhiheng Li[3]

[1] School of Earth Sciences and Engineering, Nanjing University, Nanjing, China
[2] University of Chinese Academy of Sciences, Beijing, China
[3] Key Laboratory of Vertebrate Evolution and Human Origins of Chinese Academy of Sciences, Institute of Vertebrate Paleontology and Paleoanthropology, Chinese Academy of Sciences, Beijing, China
[4] Beijing Museum of Natural History, Beijing, China

## ABSTRACT

The interplay between the pectoral module (the pectoral girdle and limbs) and the pelvic module (the pelvic girdle and limbs) plays a key role in shaping avian evolution, but prior empirical studies on trait covariation between the two modules are limited. Here we empirically test whether (size-corrected) sternal keel length and ilium length are correlated during avian evolution using phylogenetic comparative methods. Our analyses on extant birds and Mesozoic birds both recover a significantly positive correlation. The results provide new evidence regarding the integration between the pelvic and pectoral modules. The correlated evolution of sternal keel length and ilium length may serve as a mechanism to cope with the effect on performance caused by a tradeoff in muscle mass between the pectoral and pelvic modules, via changing moment arms of muscles that function in flight and in terrestrial locomotion.

# INTRODUCTION

Although the pectoral module (the pectoral girdle and limbs) and the pelvic module (the pelvic girdle and limbs) of birds are specialized for different functions, they are likely to be linked during evolution (*Allen et al., 2013*; *Gatesy & Dial, 1996*; *Heers & Dial, 2015*). This linkage could be a result of developmental and functional constraints (*Allen et al., 2013*; *Young, Hallgrímsson & Janis, 2005*), as the pectoral and pelvic limbs share a broad range of development pathways, though they acquire distinct identity in adults in tetrapods (*Young, Hallgrímsson & Janis, 2005*). Restricted by overall resources availability, pectoral and pelvic modules are negatively correlated in skeletal mass and muscle mass (*Heers & Dial, 2015*). In addition to simple resource partitioning, changes to one of the two modules, for example, an elongation of the forelimb, have implications for shifts in the position of center of mass, which can further alter the hindlimb posture and functions (*Allen et al., 2013*; *Dececchi & Larsson, 2013*; *Hutchinson & Allen, 2009*). But the functional specialization could also weaken the integration between the pectoral and pelvic limbs, as suggested by morphometric analyses of avian and mammalian limbs (*Bell, Andres & Goswami, 2011*; *Schmidt & Fischer, 2009*; *Young, Hallgrímsson & Janis, 2005*). This conflict

Corresponding authors
Tao Zhao, zhaotao@smail.nju.edu.cn, zhaotao_nju@126.com
Zhiheng Li, lizhiheng@ivpp.ac.cn

between drivers of limb evolution necessitates empirical studies to understand whether and how traits of pectoral and pelvic modules co-vary.

Along the theropod to avian lineage leading to the origin of crown birds, a series of morphological changes in the pectoral and pelvic girdles have previously been identified (*Brusatte Stephen, O'Connor Jingmai & Jarvis Erich, 2015*; *Makovicky & Zanno, 2011*). In the pectoral girdle, the changes include the enlargement of the sternum and keel (*O'Connor et al., 2015*; *Zheng et al., 2014*; *Zheng et al., 2012*), the elongation of the coracoid (*Zheng et al., 2014*), the origin of an acrocoracoid process and the triosseal canal (*Baier, Gatesy & Jenkins, 2007*; *Longrich, 2009*), the reorientation of the glenoid fossa from laterally directed to dorsolaterally directed (*Jenkins, 1993*), and the transformation of the furcula from boomerang-shaped to U-shaped (*Nesbitt et al., 2009*; *Zhou & Zhang, 2002*). In the pelvic girdle we find the elongation of the ilium and the loss of the pubic symphysis (*Hutchinson, 2001*). Of these changes, two major derived features that characterize derived birds are the larger sternal keel and the longer ilium (*Hutchinson, 2001*; *O'Connor et al., 2015*). This pattern of similar first appearances of these two key features could result from the correlated evolution between the sternal keel and the ilium, since pectoral and pelvic modules are suggested to be integrated in evolution (*Allen et al., 2013*; *Heers & Dial, 2015*). Here we compile morphometric data on extant birds and Mesozoic birds to empirically test this hypothesis based on sternal keel length and ilium length.

## MATERIAL AND METHODS

### Data collection on extant birds

We sampled 224 skeleton specimens with body mass data of 137 volant bird species from 45 families of 19 orders. All the specimens are housed in the collection of Beijing Museum of Natural History (Table S1). Sternal keel length and ilium length were taken with a digital caliper ($\pm$0.01 mm) (Fig. 1). When multiple specimens were measured for a species, the mean values of those specimens were used. These variables were log10-transformed before subsequent analyses.

### Phylogenetic comparative methods

All analyses were carried out in R 3.3.3 (*R Core Team, 2017*) using packages "ape" (*Paradis, Claude & Strimmer, 2004*), "phytools" (*Revell, 2012*) and "paleotree" (*Bapst, 2012*). Figure 2 was created using "ggplot2" (*Wickham, 2009*) and RColorBrewer (*Neuwirth, 2014*).

### Phylogeny and size-correction

We used 1,000 time-calibrated phylogenetic trees for the 137 species included in our study from birdtree.org (*Jetz et al., 2012*). Phylogenetic size-correction of log10-transformed ilium length and keel length was conducted using the function phyl.resid in the "phytools" (*Revell, 2012*).

### Evolutionary rate matrix

Under the assumption of Brownian motion model, the variance of a trait at a given time interval is equal to the length of the time interval times the Brownian motion rate parameter, $\sigma^2$. The multivariate Brownian motion is governed by the evolutionary rate matrix, which

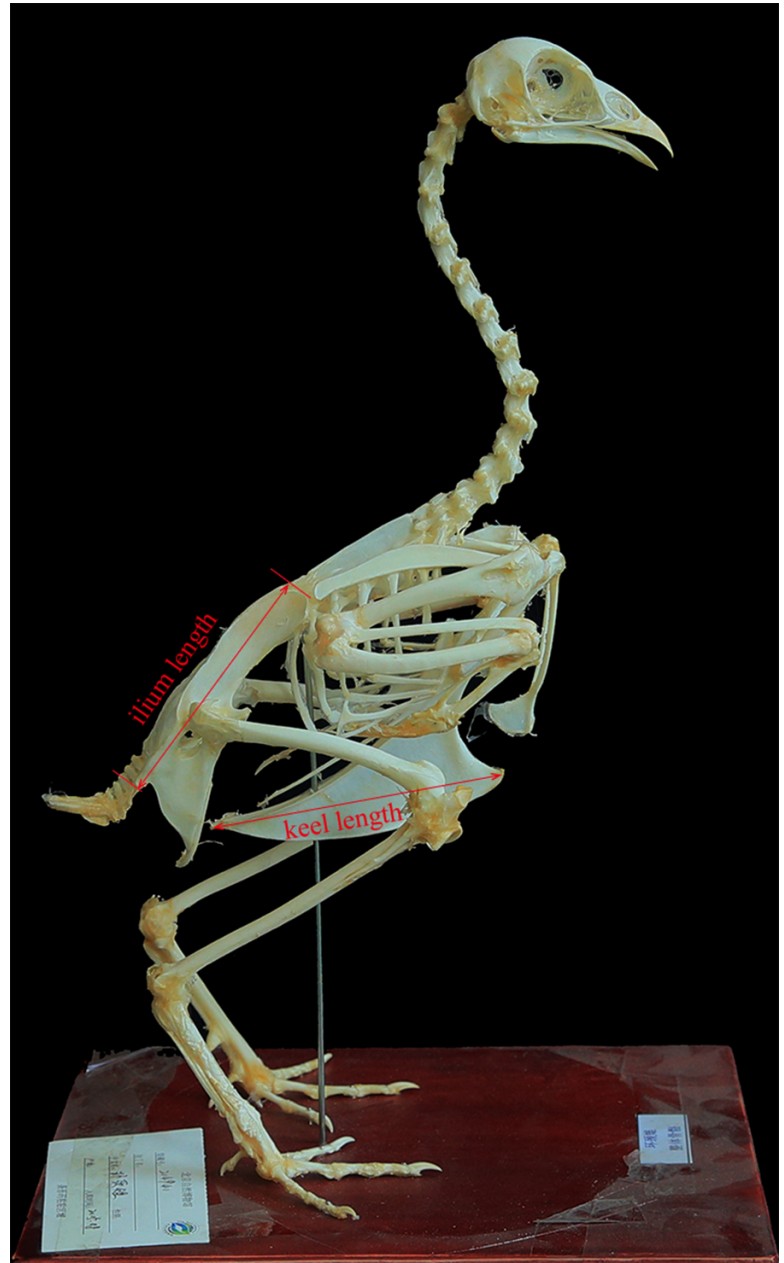

**Figure 1   Measurements used in this study (*Phasianus colchicus*, BMNH 214941, in lateral view).**
Photo credit: Qiong Wang.

contains the evolutionary variances or rates ($\sigma^2$) for individual characters on its diagonals and the evolutionary covariances on its off-diagonals (*Revell & Collar, 2009*; *Revell & Harmon, 2008*). The Pearson correlation coefficient ($r$) can be calculated based on these values. This analysis was implemented using the function evol.vcv in the "phytools" (*Revell, 2012*). The Pearson correlation coefficients from iterations across the 1,000 trees were averaged, weighted by their Akaike weights based on AICc (*Burnham & Anderson, 2002*).
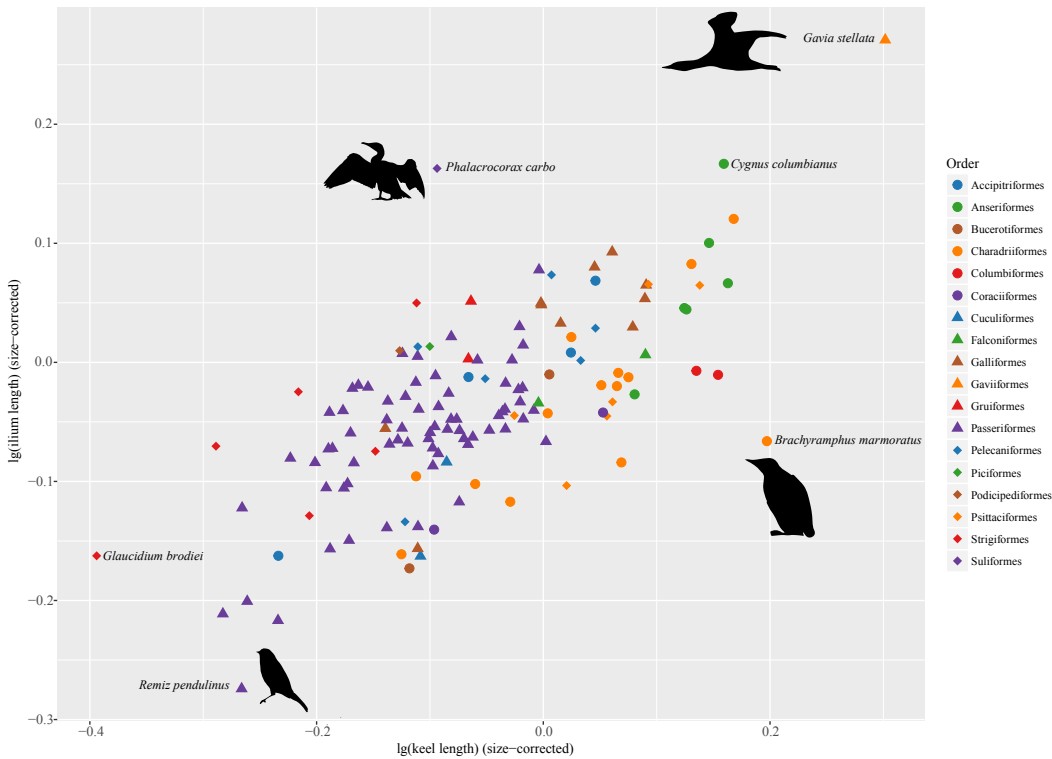

**Figure 2** **Morphospace defined by sternal keel length and ilium length showing distribution of extant birds.** Silhouettes were modified from images licensed under creative commons: *Gavia stellata* (Tony Morris, https://www.flickr.com/photos/tonymorris/429265757/); *Phalacrocorax carbo* (Tony Morris, https://www.flickr.com/photos/tonymorris/6102041629/); *Remiz pendulinus* (Michele Lamberti, https://www.flickr.com/photos/60740813@N04/8360911825/); *Brachyramphus marmoratus* (J. J. Audubon, http://www.faculty.ucr.edu/~legneref/birds/jpg/avex178.jpg).

As the Pearson correlation coefficient does not follow a normal distribution, Fisher transformation was used during the process.

## Mesozoic birds

To determine whether keel length and ilium length are correlated during early evolution of birds, we sampled 10 Mesozoic avian species housed in the collection of Institute of Vertebrate Paleontology and Paleoanthropology, Chinese Academy of Sciences, Beijing, China. Sternal keel length, ilium length and femur length were measured (Table S1). They were log10-transformed before subsequent analyses. Calibration dates for these taxa were adapted from *Wang & Lloyd (2016a)* and *Wang & Lloyd (2016b)*. A phylogenetic tree including these 10 species was constructed manually based on a recent phylogenetic analysis (*Wang & Zhou, 2017*). The fossil bird tree was time-calibrated using the function timePaleoPhy with the "equal" method in the "paleotree" (*Bapst, 2012*), with tip dates drawn randomly from a uniform distribution between the maximum and minimum dates, producing 1,000 trees. The estimate of the evolutionary rate matrix was iterated across these 1,000 trees to account for the uncertainty in time-calibration. The estimated correlation coefficients from 1,000 iterations were averaged, weighted by Akaike weights.

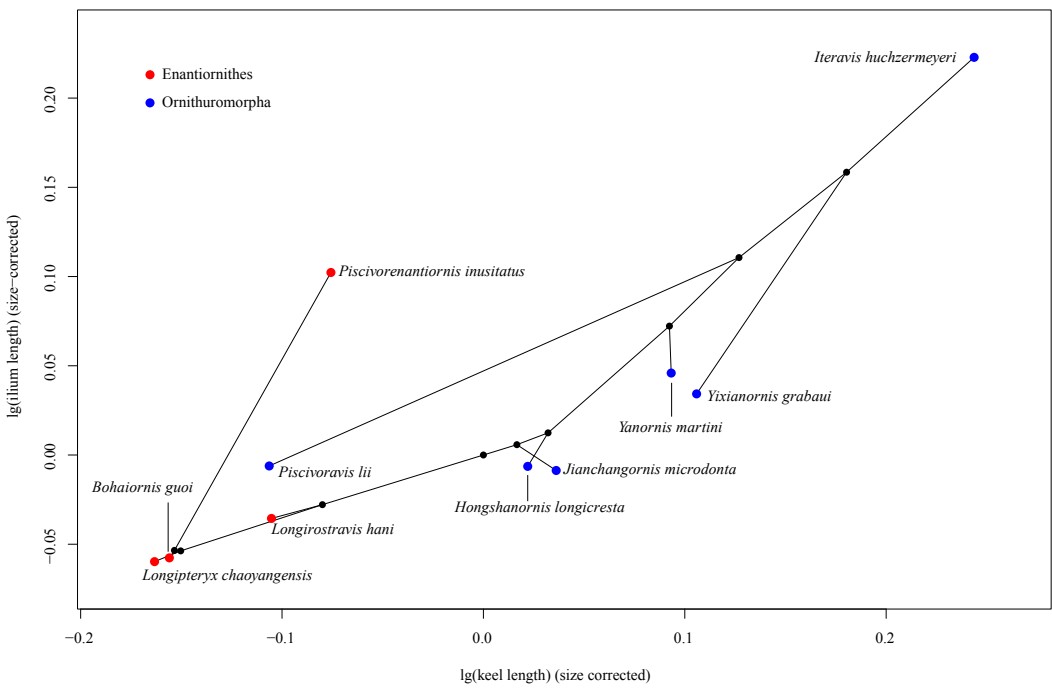

**Figure 3** Phylomorphospace depicting a Mesozoic bird tree in shape space defined by sternal keel length and ilium length.

## RESULTS

In extant birds, the correlation between sternal keel length and ilium length is 0.77 (95% CI [0.69–0.84]). Similarly, the correlation is 0.90 in Mesozoic birds (95% CI [0.61–0.98]). Both are positive and statistically significant, as their 95% confidence intervals do not include 0.

In the morphospace defined by sternal keel length and ilium length (Fig. 2), several outliers are identifiable in these extant birds. *Phalacrocorax carbo* deviates from other taxa by entering the upper-left space, indicating that it has relatively long ilia but a relatively short keel. By contrast, *Brachyramphus marmoratus* enters the lower right space, by having a relatively long keel but relatively short ilia. *Gavia stellata* also deviates from others, but it largely follows the pattern of a positive correlation between sternal keel length and ilium length.

In the phylomorphospace defined by sternal keel length and ilium length of Mesozoic birds (Fig. 3), the enantiornithines are located in the lower left part, while the ornithuromorphs in the upper right part, indicating that the ornithuromorphs have a longer keel and longer ilia than the contemporary enantiornithines. An exception is a recent described enantiornithine bird, *Piscivorenantiornis inusitatus*, which has relatively longer ilia than most ornithuromorphs except *Iteravis huchzermeyeri*. *Piscivoravis lii* differs from other ornithuromorphs in having a comparatively shorter keel and shorter ilia.

## DISCUSSION

Our results support the hypothesis that ilium length and sternal keel length are correlated during avian evolution and further provide quantitative support of the integration between pelvic and pectoral modules (*Allen et al., 2013*; *Gatesy & Dial, 1996*; *Heers & Dial, 2015*). Among basal birds, an ossified sternal keel is absent in *Archaeopteryx*, *Jeholornis* and *Sapeornis*, and only a faint keel is present in *Confuciusornis* (*Chiappe, Ji & Ji, 1999*; *O'Connor et al., 2015*; *Zheng et al., 2014*). The keel is small and restricted to the caudal part of the sternum in Early Cretaceous enantiornithines (*O'Connor et al., 2011*; *Wang & Zhou, 2017*; *Zheng et al., 2012*), while comparatively larger in ornithuromorphs (e.g., *Zhou & Zhang, 2001*; *Zhou & Zhang, 2006*). Despite these differences, the recovered positive correlation between the sternal keel length and ilium length based on data of enantiornithines and ornithuromorphs suggests that this pattern appears very early in avian evolution.

*Heers & Dial (2015)* showed that the pectoral and pelvic modules are negatively correlated in muscle mass and skeletal mass and suggested the tradeoff in investment is associated with a tradeoff in performance. In other words, the less-invested module has to cope with a larger burden. The correlated evolution of sternal keel length and ilium length may serve as a mechanism to offset, to some extent, the effect on performance caused by the tradeoff in muscle mass via changing moment arms of pectoral muscles and hindlimb muscles, because the torque produced by a muscle is determined by its mass and moment arm and the effect caused by a decrease in the muscle mass can be offset by an increase in the muscle moment arm. This requires that the mass and moment arm of a muscle can be modified independently to some extent. The sternal keel provides a surface for the attachment of muscles essential for flight, i.e., m. supracoracoideus and m. pectoralis; therefore, their moment arms can be directly affected by changes of sternal keel length. Though sternal keel length is correlated with the mass of these muscles ($R^2 = 0.47$; *Wright, Steadman & Witt, 2016*), parts of their variances cannot be statistically explained by each other. These facts imply that during evolution of flight, birds have the potential to modify masses and moment arms of pectoral muscles independently. Indeed, long-distance migratory birds can adjust the mass of pectoral muscles during their lifetime (*Dietz et al., 2007*; *Lindstrom et al., 2000*). Similarly, evolution of hindlimb functions may be achieved through changing the masses or moment arms of hindlimb muscles, though their relationship has not been empirically estimated. These inferences need to be tested in future studies.

In the sampled extant birds, two birds, i.e., *Brachyramphus marmoratus* and *Phalacrocorax carbo*, are major outliers from other taxa in the morphospace defined by sternal keel length and ilium length (Fig. 2). As a wing-propelled diver, *Brachyramphus marmoratus* has an elongated keel which accommodates the enlarged m. supracoracoideus and the elongated m. pectoralis to flap the wing in the water, which is about 800 times as dense as air (*Kovacs & Meyers, 2000*; *Spear & Ainley, 1997*). To adapt to this situation, the pelvic girdle of *B. marmoratus* shifts to an upright posture rather than acquires an elongated ilium as in other birds (Fig. 2) (*Storer, 1945*). The relatively long ilium in *Phalacrocorax carbo* is an adaptation of foot-propelled diving (*Hinić-Frlog & Motani, 2010*). Its comparatively

shorter sternal keel than that of other foot-propelled divers, for example, *Gavia stellata*, is associated with its weak flight ability; it can only slope soar in strong winds (*Norberg, 1990*). *Phalacrocorax carbo* is an example of the evolution towards flightlessness with the pelvic module enhanced and the pectoral module reduced (*Wright, Steadman & Witt, 2016*), which is seen in some flightless birds such as the Galápagos cormorant (*Phalacrocorax harrisi*) (*Livezey, 1992*) and ratites (*Cracraft, 1974*).

Among our sampled Mesozoic birds, *Piscivorenantiornis inusitatus*, a fish-eating enantiornithine (*Wang & Zhou, 2017*; *Wang, Zhou & Sullivan, 2016*), differs from other enantiornithines (*Longipteryx chaoyangensis*, *Bohaiornis guoi* and *Longirostravis hani*) in that it has relatively longer ilia (Fig. 3). The functional significance of this feature in *P. inusitatus* is unclear, but in extant birds it is associated with an aquatic lifestyle (*Hinić-Frlog & Motani, 2010*; *Stoessel, Kilbourne & Fischer, 2013*). This provides additional evidence of its ecology besides the pellet found associated with the holotype skeleton (*Wang, Zhou & Sullivan, 2016*).

In summary, pectoral and pelvic modules are linked in a more complicated way than just negatively correlated in overall investment. Besides modifying moment arms of muscles, birds may change behaviors to cope with the effect caused by tradeoff in investment. Moreover, these two modules may be linked through avian eggs, the shape of which is suggested to be correlated with both the pelvic shape (*Dyke & Kaiser, 2010*; *Mayr, 2017*) and flight ability (*Stoddard et al., 2017*). More integrative studies in the future can provide more insight into the relationship between pectoral and pelvic modules.

## ACKNOWLEDGEMENTS

We thank Mr. Zhaohui Zeng for access to specimens housed in Beijing Museum of Natural History and Mr. Qiong Wang for taking the photo for Fig. 1. Comments from Jonathan Mitchell and T. Alexander Dececchi improved the manuscript.

### Funding

The research was supported by the National Natural Science Foundation of China (91514302, 41688103). The funders had no role in study design, data collection and analysis, decision to publish, or preparation of the manuscript.

### Grant Disclosures

The following grant information was disclosed by the authors:
National Natural Science Foundation of China: 91514302, 41688103.

### Competing Interests

The authors declare there are no competing interests.

## Author Contributions

- Tao Zhao conceived and designed the experiments, performed the experiments, analyzed the data, contributed reagents/materials/analysis tools, wrote the paper, prepared figures and/or tables, reviewed drafts of the paper.
- Di Liu performed the experiments, contributed reagents/materials/analysis tools, prepared figures and/or tables, reviewed drafts of the paper.
- Zhiheng Li performed the experiments, wrote the paper, reviewed drafts of the paper.

## Data Availability

The raw data and code have been provided as Supplemental Files.

## Supplemental Information

Supplemental information for this article can be found online at http://dx.doi.org/10.7717/peerj.3622#supplemental-information.

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
