# Peer review of "Correlated evolution of sternal keel length and ilium length in birds"

_PeerJ, doi:10.7717/peerj.3622_

## Round 0.1 · original submission · Minor Revisions

You have received two very detailed reviews that will help you to greatly improve your manuscript. Please carefully take into account the opinion of the reviewers and answer point by point their suggestions and objections.

·

Basic reporting

The text is well-written and the citations are solid. See General comments for further remarks.

Experimental design

The research question is well defined and the data are robust. See general comments for my (minor) methodological description concerns.

Validity of the findings

The two key results are well demonstrated and supported by the data. Again, see general comments for my (again minor) concerns.

Additional comments

Regarding the paper as a whole: This papers makes a valuable contribution in outlining a connection between two locomotor modules in birds. The evolution of the pectoral and pelvic modules has long been postulated as integral to the success of birds and their ecological radiation, and this new study robustly outlines an evolutionary link that has gone largely unexplored. Below, I outline several general concerns, in descending order of importance, but one thing I would like to emphasize first is that the Brownian motion model (and correlations derived from it) should be taken with a slightly larger grain of salt. If, in one ecological category, two recently-diverged taxa have very different trait values, the entire rate estimate for that category will skyrocket. If all ecological categories are robustly defined and comparable, this is somewhat less of an issue. But in some cases, such as Brachyrhamphus (a diving bird that mostly swims) and the Larus clade (gulls which are rather ecologically generalist), such a divergence can have a large impact due solely to the fact that the "swimming" category encompasses a wide range of ecological behavior, and thus a wide range of expected morphologies. A few cases where sister clades diverge a lot in both morphology and ecology (but retain identical categorizations) can have a tremendous impact on reconstructed rates and the Akaike fit of different models. I point this out only to encourage the authors to reframe certain aspects as a bit more circumspect (e.g., is the difference between r = 0.71 and r = 0.66 really that dramatic given uncertainties in ecological classification & the nature of the model?). I think the core results of 1) a connection in illial and sternal keel length is well-documented, and 2) the variance of that relationship among ecological groups are well-supported and interesting.

Regarding the Data description: I think this paper would be greatly helped by a bit more explicit discussion of the ecological categories and how birds were separated into them. For instance, in the "swimming" group are the Larus gulls, which spend a lot of time walking on the ground, flying and (yes) swimming. But also Grebes like Tachybaptus which spend almost no time at all walking around on the ground. Likewise, the "terrestrial" group includes birds that almost never touch the ground (e.g., Hirundo swallows) and birds that almost never leave the ground (e.g., Coturnix quails). I do not think there's anything wrong with the authors' classifications, and too-finely dividing the categories would be difficult with only 137 species in the dataset, but a more frank discussion of the rationale and meaning of each group would greatly assist readers in assessing the meaning behind their data.

Regarding Akaike Weights: Support for the arguments in this paper is quantified using the Akaike Information Criterion. This is certainly a sensible way to do things, and it's become something of the standard in phylogenetic comparative methods, so I have no problem with the authors' choice here. However, I am a bit confused as to what, exactly, they did. As I point out in the specific comments below, on line 115 they refer to a "sum" of Akaike weights, and I'm not sure I fully grasp why. Further, in the code itself, the line where they calculate AICc implies that their Brownian motion model has 17 parameters (!), and I can't discern what those 17 parameters are, or why they're calculating AICc in that way. More clarity in explaining precisely what the authors did would be very helpful, especially given my comment below on their code.

Regarding the code: Everything *looks* fine in the code itself, but I have some very minor quibbles. The authors provide the raw data in an Excel file, which is great! But it'd be nice if they also provided a summary file. Basically, running the entire analysis on 1000 trees requires more computation time than I can spare for a review. It'd be nice if they broke the code into two parts: Lines 1-85, and lines 85-the end, and the "RR" list were saved as RR.RData (using the save() function in R). That would increase reproducability by allowing folks to directly query the elements of the RR object, and understand all subsequent analyses of said object, without running the code above that line.


Specific Comments:
Lines 31-32: Positive correlation between keel/illium based on "requirement ot counterbalance the enlargement of pectoral module" a bit speculative.

Line 72: Slightly more detail is needed here. If you measured 5 specimens from a particular species (say, 3 male & 2 female), did you just take the mean value of those 5 specimens? Or did you take a male average, a female average, and then an average of those two averages?

Line 115: The "sum" of Akaike weights? I'm not 100% clear on what was done here. I'd assume that, for each tree, you compared the fit of the one-matrix and five-matrices models using Akaike weights (so that, e.g., for tree 1 you'd have weights of 0.1 and 0.9). Then I'd imagine you would determine 1) frequency of Akaike Wt_{5-matrices} > Akaike Wt{1-matrix} and 2) the average of Akaike Wt_{5-matrices} across the 1000 trees. Until this line, that's what I was assuming. But if "the sum of Akaike weights" is 0.94, then the average Akaike weight would have to be 0.0094, which is terrible. This is my long way of saying that I assume the "sum" here is meant to be "mean", but that the description of the methods employed is unclear enough that I can't be certain.

Line 127 - 129: There seems to be a lot to unpack in this sentence. Specifically, the distinction of a "strong" correlation in ground birds (r = 0.71) and "moderate" in swimming bird (0.66). The authors do their readers a great service by laying all of these correlations out, but a slight disservice by adding the strong and moderate designators. It's also incredibly unclear to me what these correlations actually are. I would assume that they're the mean correlation across the 1000 trees. Is that accurate? If so, why are there no standard deviations reported alongside them?

Lines 148 - 154: This argument seems to rely on the assertion that there is greater "relaxtion of selective pressure to maintain the flight ability" in ground birds over wading birds. This may be true, but Rails are typically classified as wading birds, and they routinely lose their ability to fly, which would suggest that selection for flight is pretty relaxed in at least some wading bird groups as well.

Lines 158 - 160: The "higher rates" in swimming birds seem to be driven almost entirely by two species pairs. Based on Figure 3, the sister pair at about (-0.15, 0.18) & (0.25, 0.25) and the clade centered around (-0.18, -0.13) & its sister species at about (0.2, -0.1). What are these birds? These two nodes seem to be driving much of the higher rate and lower correlation in swimming birds, and the way Brownian motion models work, these two clades have an outsized impact on the whole analysis. More detail on them would be fantastic.

Lines 162 - 183: I certainly agree with the authors about the fact that skeletal correlates, like illium length, improve our ability to reconstruct life habits in fossil organisms. However, given that the dataset includes no flightless birds and so few weak fliers, that it's hard to extrapolate the results down the tree.

Code-line 34: It's safer to add a "library(parallel)" call here, too. This is necessary in some versions of R for the detectCores() function call.

·

Basic reporting

I believe this article passes this criteria, though I do suggest the authors add a few references (see the general comments) to more fully capture the bod of knowledge that exists on this topic. The figures and tables are clear and professional and the article is self contained.

Experimental design

No comment

Validity of the findings

I have some issues, as raised in my general comments, on some aspects of the results and the speculation on how they can be extended to early avians. Taken as a whole the findings of this study are sound.

Additional comments

While I enjoyed reading this manuscript and am intrigued by the authors findings (and the implications) I do have some comments I believe will help improve this work going forward.

Line 47: Perhaps you should include Dececchi and Larsson 2013 which specifically examines this issue across the theropod line into birds. Also perhaps listing other references on limb function pre-flight and its effect on the origin of birds may be of use here, may I suggest Hutchinson and Allen 2009 as a good general review of the topic.

Line 53: This is a very broad statement, maybe more clearly identifying some of the major differences (changes in coracoid shape, rotation of glenoid fossa, formation of the triosseum canal, furcular shape and proportions etc) that are suggested in these and other references would make it clearer for non-experts reading this paper. 

Line 61: There is a lot of developmental work on this topic that should be brought up here as well including work on limb feathers (Domyan et al. 2016) that show how the two systems are linked. Also there has been work in the mammalian systems which could be mentioned Schmidt and Fischer (2008). That said you should also mention that there are studies that suggest that the link between fore and hindlimb may not be that strong, specifically in birds (see Bell et al. 2011). 

Line 83: I understand the need to differentiate taxa into categories but would be cautious on these ones, especially as in reality they are not very clear cut. Glen and Bennett 2007 discuss this in depth and talk about how your “terrestrial” category includes at least 3 separate subcategories that more accurately reflect the lift history of many bird groups. Looking at your taxon list I can see several taxa that are segregated into one category which inhabit more than one group (for example Corvus and the Turdus species forage extensively on the ground but perch and roost in the trees). Perhaps just mentioning how taxa inhabit more than one category and maybe in the discussion mention how this could influence the results would be sufficient.  

Line 143: Can you match your taxa up with myological data (i.e. keel length with pectoral mass and ilium with pelvic limb) to confirm? I bring this up because there is a lot of variation in Heers and Dial's dataset. Looking at it I can see among birds of prey the Forelimb/ hindlimb muscle mass ratio varies from 1.12 to 4.75. A lot of this is variation in the hindlimb (in that same dataset it varies from 4.91% to 23.5% while the forelimbs only vary between 19.9 and 31.5% among birds of prey) so linking your morphometric data to muscle mass measurements for those specific taxa would make this case stronger.

Line 145: Should not swimming birds require high torque values as well, especially as often they spend significant amounts of time on land and the hindlimb is the main propulsive unit it the water? Also in your classification you have gulls and ducks, which paddle along the surface, in the same class as loons and cormorants which propel themselves underwater using their feet. These behaviours are very different and put different demands on the hindlimbs, as mentioned earlier having such broad categorizations may be hiding important evolutionary information. 

Line 151: But many of the "terrestrial" taxa are strong fliers (such as the corvids, the tree sparrow, the black bird and starlings) so this selective pressure is not relaxed. It again points to the almost arbitrary nature of the “terrestrial” versus “arboreal” as all these birds can and do perch frequently and many “arboreal” birds do feed extensively on the ground.

Line 153: This is, in the reviewers mind, a significant limitation on the predictive ability of this study. I suggest that the authors add a few flightless taxa to test their hypothesis it would elevate this work. 

Line 160: You have not taken into account the differences in flight styles among the "swimmers". Gulls glide a lot during flight compared to loons and ducks who use fast wing beats (see Viscor and Fuster 1987). This has led to significant variation in muscle masses and wing shapes between these taxa that make this a particularly diffuse category. This could be effecting your signal.

Line 168: I would argue that given the fact that all the listed taxa are known form multiple specimens from lagerstatten deposits that frequently preserve soft tissue means that if present any cartilaginous sternal keel should be present in some specimens. We have a very poor fossil record of early birds outside these types of deposits so I would suggest that missing data due to preservation is not a major issue in this context.

Line 173: I would be cautious. The earliest birds likely shared hindlimb function and muscle proportionality with closely related maniraptoran theropods (including posture, muscle moment arms etc) than later avians till around Pygostylia. Among non-avian theropods we do not see the same pattern in the limbs (increased forelimb linked to decreased hindlimb dominance). If you look at the dataset from Allen et al. 2013 you see a significantly reduced pattern of increasing forelimb muscle proportion means decreasing hindlimb muscle proportion (for example Coelophysis, Tyrannosaurus and Archaeopteryx have the same hindlimb percentages with vastly different forelimbs). Also per Dececchi and Larsson 2013 we find that Microrptorines have exceptionally long hindlimbs as well as long forelimbs and that the divergent pattern between limb lengths does not come in until well after the node Aves.

Line 178: This can be test rather simply and I think the authors should do that. We know a lot about the pattern of the keel's evolution, how non-linear it is, in fact Mayr 2017 discusses this in great length. It would be fairly simple for the authors to get a sample of early avians with both a preserved keel and Ilium to test this hypothesis. I believe it would increase the impact of this manuscript to do so.  

Line 183: I would suggest caution. These results indicate that ecological guild influences this pattern, and the guild (especially as it concerns arboreal versus terrestrial) of early birds and their ancestors is still a subject of debate (see for example Dececchi and Larsson 2011 or Bell et al. 2011) as well as the fact that stemwards of Ornithuromorpha (especially below Pygostylia) there are good reasons to believe this relationship may not hold, or at least the scaling factors will be very different.

---

## Round 0.2 · Minor Revisions

As you may see we are nearly there. Your manuscript needs some minor, cosmetic modifications. Please, do them as soon as possible, I am ready to accept it.

·

Basic reporting

The paper is clear and well-written, although I suggest a few minor sentence alterations below.

Experimental design

The design of the analysis is robust and interesting. The question concerning the association between different skeletal elements adds new data.

Validity of the findings

The findings are well supported by the data.

Additional comments

This paper presents the same interesting and informative correlation of skeletal measures in birds that it did before, but improves upon it greatly by incorporating better coverage of fossil data and an even more intriguing discussion. I believe this paper is a worthwhile addition to our understanding of skeletal morphology in birds, and should be published. Below I have a few (very minor) suggestions for improving the quality of the manuscript.

Figure 1: I think it would be better to use a color other than white for the sternal and ilial measurements and words. They kind of blend into the bone, as it is. Maybe red?

Figure 2: This is a very nice figure, and it doesn't *need* any improvement. However, I'd recommend that the authors look at R Color Brewer for color choice, as it will make the figure look even nicer. Color Brewer is both an R package and is available as a website for color choice (http://colorbrewer2.org/)

Line 84: Cite the R packages used here (e.g., ape, phytools)

Line 169: Replace "exhibit huge deviation" with "are major outliers"

Line 186: Replace "a side evidence" with "additional evidence"

·

Basic reporting

No comment

Experimental design

no comment

Validity of the findings

No comment

Additional comments

I applaud the authors for listening to both reviewers comments and implementing them. I believe it has improved an already quite strong manuscript.

I only have some minor comments, mostly to do with wording but nothing of major substance that should delay acceptance of this manuscript. I have attached my comments to the marked up version of this manuscript to help streamline the process of incorporating any of my suggestions into the final product.

Sincerely Dr. TA Dececchi

---

## Round 0.3 · accepted · Accept

Thank you for your consideration of the detailed reviews. I think that we are done. Nice manuscript.